# The Association between Dietary Protein Intake and Sources and the Rate of Longitudinal Changes in Brain Structure

**DOI:** 10.3390/nu16091284

**Published:** 2024-04-25

**Authors:** Fusheng Cui, Huihui Li, Yi Cao, Weijing Wang, Dongfeng Zhang

**Affiliations:** 1Department of Epidemiology and Health Statistics, Public Health College, Qingdao University, Qingdao 266021, China; qducfs19981019@163.com (F.C.); li1772650040@163.com (H.L.); zhangdf1961@126.com (D.Z.); 2Biomedical Center, Qingdao University, Qingdao 266021, China

**Keywords:** dietary protein, animal protein, vegetable protein, sources of protein, brain structure markers, UK Biobank

## Abstract

Few studies have examined dietary protein intake and sources, in combination with longitudinal changes in brain structure markers. Our study aimed to examine the association between dietary protein intake and different sources of dietary protein, with the longitudinal rate of change in brain structural markers. A total of 2723 and 2679 participants from the UK Biobank were separately included in the analysis. The relative and absolute amounts of dietary protein intake were calculated using a 24 h dietary recall questionnaire. The longitudinal change rates of brain structural biomarkers were computed using two waves of brain imaging data. The average interval between the assessments was three years. We utilized multiple linear regression to examine the association between dietary protein and different sources and the longitudinal changes in brain structural biomarkers. Restrictive cubic splines were used to explore nonlinear relationships, and stratified and sensitivity analyses were conducted. Increasing the proportion of animal protein in dietary protein intake was associated with a slower reduction in the total hippocampus volume (THV, *β*: 0.02524, *p* < 0.05), left hippocampus volume (LHV, *β*: 0.02435, *p* < 0.01) and right hippocampus volume (RHV, *β*: 0.02544, *p* < 0.05). A higher intake of animal protein relative to plant protein was linked to a lower atrophy rate in the THV (*β*: 0.01249, *p* < 0.05) and LHV (*β*: 0.01173, *p* < 0.05) and RHV (*β*: 0.01193, *p* < 0.05). Individuals with a higher intake of seafood exhibited a higher longitudinal rate of change in the HV compared to those that did not consume seafood (THV, *β*: 0.004514; *p* < 0.05; RHV, *β*: 0.005527, *p* < 0.05). In the subgroup and sensitivity analyses, there were no significant alterations. A moderate increase in an individual’s intake and the proportion of animal protein in their diet, especially from seafood, is associated with a lower atrophy rate in the hippocampus volume.

## 1. Introduction

As life expectancy continues to rise, the prevalence of brain aging and cognitive-related diseases is increasing [1,2,3]. Aging often accompanies changes in brain structure markers, which may contribute to the development of cognitive-related diseases [4]. Existing research has indicated that longitudinal changes in brain structural markers can to some extent reflect alterations in the progression of neurodegenerative diseases [5,6,7,8]. It is crucial to identify modifiable risk factors to slow down the changes in neurodegeneration-related brain structure markers.

Many studies have suggested that dietary factors, as a modifiable component of an individual’s lifestyle, play a vital role in preserving brain health [9,10,11]. Protein, as a primary component of the human diet, has garnered widespread attention in this context. Protein constitutes a fundamental component of neurotransmitters and neurons, contributing to the maintenance of brain structure and function [12]. Research has suggested that moderate dietary protein intake may reduce the brain’s amyloid-beta (A*β*) plaque burden in older adults to delay the onset of Alzheimer’s disease [13,14]. Therefore, we speculate that alterations in dietary protein intake may contribute to changes in brain structure markers.

However, few studies have examined dietary protein intake and sources, in combination with longitudinal changes in brain structure markers. Moreover, in previous studies on brain structural markers, most were cross-sectional and used a single wave of brain image data.

Therefore, we employed a cohort study conducted by the UK Biobank (UKB), utilizing two waves of brain imaging data, to examine the association between dietary protein intake and different sources of dietary protein, with the longitudinal rate of change in brain structural markers.

## 2. Materials and Methods

### 2.1. Study Population

The data used in this study were obtained from the UKB, comprising participants aged 40–69 years at the time of recruitment in 2006–2010 [15]. Over 500,000 UK residents were recruited through 22 assessment centers [16]. Following dietary assessments, the UKB initiated a multi-modal imaging sub-study [17], with nearly 50,000 participants assessed by the end of 2019. From 2018 to 2022, participants from the initial assessment were invited to undergo repeat imaging assessments. All UKB participants provided written informed consent, and ethical approval for this research was obtained from North West–Haydock Research Ethics Committee, with reference number 16/NW/0274. The current study was conducted under application number 95715, utilizing the resources provided by the UKB accessed on 10 July 2023. Further details can be found on the UKB website (https://www.ukbiobank.ac.uk/enable-your-research/apply-for-access).

### 2.2. Assessment of Dietary Protein Intake and Sources

Dietary information was obtained from a 24 h dietary recall questionnaire. Participants in the UKB took part in 5 waves of online surveys during this period (2009–2012). Using the acquired dietary data in the UKB, the daily protein intake (g/d) for each participant was calculated, including both plant and animal protein [18,19]. In addition, we calculated the ratio of animal protein to vegetable protein in relation to the total protein intake. We also compared the consumption of animal protein to vegetable protein. These calculations were made to provide insights into the participants’ dietary protein levels.

In addition, based on previous studies on protein sources in European diets [20,21], we adjusted the classification of dietary protein sources. We categorized them into 8 types of animal protein sources and 3 types of plant protein sources. Detailed information and the codes regarding food protein sources can be found in Appendix A.

### 2.3. The Rate of Change in Brain Structural Markers

MRI data were acquired during the third (2014+) and fourth assessment (2019+) visits at three imaging centers, equipped with identical scanners (Siemens Skyra 3T running VD13A SP4 with a Siemens 32-channel RF receive head coil, Munich, Germany). The average interval between the assessments was 3 years. Structural magnetic resonance (MR) imaging was utilized to estimate the total brain volume (TBV), gray matter volume (GMV), white matter volume (WMV), hippocampus volume (HV) and white matter high-intensity volume (WMHV). The MR imaging protocols have been detailed elsewhere [17]. All the information on the structural image segmentation and data normalization is available elsewhere [22]. Publicly available image processing tools, primarily from the FMRIB Software Library, were employed for data processing, utilizing the output of the standard biobank processing pipeline. All data were normalized for head size. Additionally, a new variable, representing the rate of change in brain structure markers, was calculated to depict the longitudinal changes in the brain structure of the participants. A smaller value of the rate of change between the two measurements indicates a faster decrease in brain volume.

The rate of longitudinal change in brain structural markers = [brain image data (2019+) − brain image data (2014+)]/brain image data (2014+).

### 2.4. Covariates

In our study, certain confounding factors were adjusted for [23,24]. Demographic characteristics were collected at recruitment, including age, sex, energy, ethnicity, education and Townsend deprivation index (TDI). The TDI represents the social deprivation status and was categorized as low, medium or high deprivation. Participants’ education was categorized as college, above or below. Physical activity, smoking, alcohol consumption and body mass index (BMI) were adjusted for, as lifestyle factors. The BMI of the participants was categorized as underweight (BMI < 18.5), normal weight (BMI ≥ 18.5 but <25.0) and overweight/obese (BMI ≥ 25.0). Participants were grouped into low, moderate and high activity levels based on metabolic equivalent minutes per week. The baseline disease status, encompassing cancer, cardiovascular diseases (CVDs), hypertension and diabetes, was determined using the participant’s electronic records. Additionally, we accounted for the polygenic risk Scores for Alzheimer’s disease (AD-PRS) to control for genetic factors.

### 2.5. Statistical Analysis

The baseline characteristics of the participants were described separately, by sex. Continuous variables were presented as the mean ± standard deviation. Categorical variables were expressed as percentages. We used a multiple linear regression model to investigate the relationship between dietary protein intake (animal protein, plant protein and total protein), the ratio of animal protein to plant protein in regard to the total protein intake, and the relative comparison of animal protein to plant protein, with changes in brain structure. Additionally, the data were transformed based on the distribution type of the variables to approximate normal or symmetrical distribution. To control for the influence of confounding factors, we established three models for adjustments. The *β* was adjusted for age and sex in model 1. Model 2 was additionally adjusted for TDI, education, energy, physical activity, smoking (ever smoked or not), alcohol intake (ever drunk or not) and body weight status. Model 3 was additionally adjusted for baseline cancer, CVDs, hypertension and diabetes. To address the issue of multicollinearity among the independent variables, we calculated the variance inflation factors (VIFs) and tolerances for collinearity diagnosis. In addition, to examine the presence of nonlinear relationships, we also introduced restricted cubic spline analysis to explore dose–response relationships in our analysis.

We further analyzed dietary protein sources and examined the associations between various dietary protein sources and changes in brain structural biomarkers using multiple linear regression, with multiple adjustments made. Participants were categorized based on the source of the dietary protein intake. Those with a dietary protein intake of 0 were classified into the none intake group, while non-zero intake levels were divided into lower and higher intake groups based on the median intake.

In sensitivity analyses, we performed subgroup analyses based on sex to explore the impact of different sexes on the results. In females, we also adjusted for oral contraceptive use, additionally [25,26,27]. Furthermore, to control for the influence of genetic factors, we additionally adjusted for AD-PRS. Additionally, we restricted the analysis to participants who had completed at least two dietary recalls. Moreover, we conducted repeated analysis that excluded baseline neuropsychiatric disorders (including depression, epilepsy and encephalitis) [28].

Statistical analyses were performed using R 4.2.3, and 2-sided *p* values < 0.05 were considered statistically significant.

## 3. Results

### 3.1. Participant Characteristics

Among the 210,948 UKB participants who had at least one dietary assessment, we excluded participants who were missing data from two waves of imaging, as well as those with incomplete dietary data (total energy = 0 MJ or ≥20 MJ [29,30]) and missing covariates. After excluding participants based on covariates, there were no baseline dementia cases. This resulted in a final analysis dataset comprising 2723 and 2679 participants, as detailed in the flowchart (Figure 1). The average follow-up time was 8 years.

Among the 2723 participants in the study on the TBV, WMV, GMV and HV, the average (SD) age was 52.66 (7.42) and 51.7% were female (Table 1). In the additional group of 2679 participants that included the WMHV, the average (SD) age was 52.7 (7.41), with 51.7% being female (Appendix A). Men were more likely than women to have a higher intake of protein, animal protein, plant protein and total energy intake. They were also more likely to suffer from cardiovascular diseases, diabetes and hypertension.

### 3.2. Dietary Protein and Brain Structure

A significant association was not observed between dietary protein intake (including animal and plant protein) and longitudinal changes in the TBV, WMV, WMHV and GMV. However, a significant association was found in terms of the HV (Table 2, Figure 2, Appendix A). In the overall adjustment model, an increasing proportion of animal protein in the dietary protein intake was associated with a slower reduction in the total hippocampus volume (THV, *β*: 0.02524, *p* < 0.05), left hippocampus volume (LHV, *β*: 0.02435, *p* < 0.01) and right hippocampus volume (RHV, *β*: 0.02544, *p* < 0.05). A higher intake of animal protein relative to plant protein was linked to a lower atrophy rate in the THV (*β*: 0.01249, *p* < 0.05) and LHV (*β*: 0.01173, *p* < 0.05) and RHV (*β*: 0.01193, *p* < 0.05). However, there was no significant association between the total dietary protein intake and the longitudinal rate of change in the hippocampus volume.

After adjusting for all factors, the longitudinal rate of change in the THV and LHV showed a significant positive correlation with the absolute intake of dietary animal protein (LHV, *β*: 1.522 × 10^−4^, *p* < 0.05; THV: *β*: 1.188 × 10^−4^, *p* < 0.05), while plant protein exhibits a negative correlation (LHV: *β*: −0.0003901, *p* < 0.05; THV: *β*: −0.000319, *p* < 0.01). However, such an association was not observed in the analysis of the right hippocampus (Table 2, Figure 2).

The results from the restricted cubic spline analysis indicated a clear nonlinear association in the right hippocampus (Figure 3). Both the proportion of animal protein to total protein and the intake of animal protein showed evident nonlinear associations with the longitudinal rate of change in the volume of the right hippocampus (*p* for overall < 0.05 and all *p* for nonlinearity < 0.05). The proportion of plant protein to total protein shows an approximately inverted “J”-shaped association with the rate of change in the hippocampal volume. However, no nonlinear associations were found between the longitudinal rate of change in other brain structure biomarkers and protein intake (*p* for overall > 0.05 and all *p* for nonlinearity > 0.05, Appendix A).

### 3.3. Dietary Protein Sources and Brain Structure

In the total hippocampus and the right hippocampus, individuals with a higher intake of seafood exhibited higher longitudinal rates of change compared to those with no consumption (THV, *β*: 0.004514; *p* < 0.05; RHV, *β*: 0.005527, *p* < 0.05), but this was not observed in individuals with lower seafood intake (*p* > 0.05). Similarly, individuals with a relatively lower intake of nuts exhibited a lower rate of atrophy in the total hippocampus compared to those who did not consume nuts, while no significant association was observed at the higher intake level. Lower cheese intake was associated with a higher longitudinal rate of change in the left hippocampus (Figure 3 and Appendix A).

### 3.4. Subgroup and Sensitivity Analyses

In males, a reduction in plant protein intake and proportion was associated with a lower rate of volume atrophy in the total hippocampus and the left hippocampus (*p* < 0.05, Appendix A). In females, after adjusting for covariates, there was a significantly positive correlation between the proportion of animal protein to total protein and the rate of change in the hippocampus volume (including total, left and right, *p* < 0.05). After adjusting for oral contraceptive use, the association still existed (Appendix A). However, a significant positive association with animal protein intake was only found in regard to the total volume of the hippocampus (Appendix A). We further adjusted for AD-PRS. The association between dietary protein intake and the longitudinal change rate of the hippocampus did not show significant changes (Appendix A). Additionally, we repeated the analysis by including participants with at least two waves of dietary data. We found that the results did not significantly alter (Appendix A). When restricting the analysis to participants without neurological disorders, the association also remained unchanged (Appendix A). The sensitivity analysis results indicated the robustness of the study findings.

## 4. Discussion

In this prospective study conducted in the UKB, we observed that an increase in animal protein intake and proportion were associated with a lower longitudinal rate of atrophy in the hippocampus volume. In the subgroups based on gender, these associations were similar, and there was no significant alteration according to the genetic risk of dementia. Additionally, higher intakes of seafood and cheeses were associated with a higher longitudinal rate of change in the hippocampus volume.

Our study did not find any association between the total protein intake and the longitudinal change rate of brain structural markers. Conversely, a higher intake of animal protein and the proportion of animal protein to the total protein intake showed a positive association with the longitudinal change rate of the hippocampus volume. Findings from previous studies suggested that participants with a higher proportion of protein in their diet had a lower risk of dementia [31,32,33]. However, in another study, it was found that patients with neurodegenerative diseases had similar protein intake compared to the control group. Seafood is a good source of dietary protein. We hypothesize that, concerning the longitudinal change rate of the hippocampus volume, the impact on brain health may not be solely related to the absolute dietary intake of protein, but is more likely influenced by altering the proportion of animal protein in the diet. The association may also be explained through dietary protein sources such as seafood. Additionally, a cross-sectional study showed a significant association between seafood consumption (at least one meal per week) and reduced pathological changes in the brain [34].

In our study, we observed a significant association between dietary animal protein and hippocampal volume changes. The hippocampus, as a complex structure in the brain, has been found to be linked to various diseases through alterations in its volume. Previous research has shown associations between hippocampal volume changes and conditions such as Alzheimer’s disease, epilepsy and encephalitis [35,36]. Additionally, individuals with depression tend to have smaller hippocampal volumes compared to healthy controls [37]. The sensitivity analysis results indicated the robustness of the study findings after excluding participants with baseline neurological and psychiatric disorders. Furthermore, findings from another cross-sectional study suggested that participants who abstain from meat in their diet have a higher incidence of depression [38]. Thus, a reduced intake of animal protein might contribute to neuropsychiatric disorders that have been previously associated with a smaller hippocampal volume. Further in-depth research and exploration are warranted.

Animal protein and plant protein exhibit significant differences in their amino acid composition [39,40,41,42]. Animal protein, especially from seafood, typically contains all the essential amino acids (EAAs). These amino acids are essential for the human body, and the composition and proportion of amino acids in animal protein are similar to those in the human body. In contrast, plant protein may sometimes lack or have lower levels of certain EAAs, such as branched-chain amino acids (BCAAs, especially leucine), tryptophan and lysine [43,44,45], making it an incomplete protein. On the other hand, plant protein often comes with some antinutrients, such as lectins, phytic acid and saponins. Furthermore, animal protein is generally more easily absorbed and utilized by the human body, with higher biological availability. In addition, low BCAA levels may impair brain structure and function [46,47]. Current research has suggested that BCAAs, particularly leucine, may serve as crucial donors of nitrogen in the brain, participating in the cycle of glutamate and glutamine [48]. Glutamate, a major excitatory neurotransmitter in the brain, regulates various functions within the brain [49,50]. Secondly, the cycle between glutamate and glutamine in the brain may be restricted due to low brain BCAA levels. This could lead to the accumulation of glutamate and ammonia in the brain, resulting in neurotoxicity and neurodegenerative changes that impair brain health [46,50]. Thirdly, reduced levels of brain BCAAs may diminish protein synthesis, impacting the repair of brain tissue, synaptic growth and remodeling [46,51]. Fourth, supplementation of branched-chain amino acids may potentially reduce the levels of neuroinflammation in the brain to protect its structure [52]. These reasons may explain our results on the association between animal protein intake and the rate of brain atrophy.

To the best of our knowledge, there has been limited exploration of the longitudinal relationship between dietary factors and markers of brain structure. We adjusted for multiple covariates to control for the influence of confounding factors and stratified the analysis by gender to examine differences between different gender groups. Furthermore, we utilized both absolute and relative indicators to evaluate the extent of dietary protein intake. Previous studies investigating the association of lifestyle or dietary factors with brain structural markers have predominantly relied on single-wave brain imaging data as outcome measures [9,11,26], introducing variability due to individual differences and the influence of different imaging centers. In contrast, our study employed two waves of brain imaging data to calculate the longitudinal change rates. This approach aimed to comprehensively understand the influence of dietary factors on the brain structure, while mitigating potential confounding factors.

However, there are still some limitations. Firstly, 24 h dietary recall may not accurately represent long-term dietary habits, and the data, being self-reported, may introduce recall bias. However, we averaged multiple waves of dietary data to reduce random errors and mitigate the impact of individual variations. Secondly, due to potential residual confounding and reverse causation, our study results may not necessarily reflect a causal relationship, even though the imaging assessment was conducted long after the dietary assessment. Thirdly, imaging examinations may have the potential to exert health-promoting effects on the participants. Lastly, in the current study, the majority of the study population consisted of white individuals. Prudence is advised when extrapolating the results to different populations.

## 5. Conclusions

In conclusion, current research findings indicate that a moderate increase in the intake and proportion of animal protein in a person’s diet, especially from seafood, is associated with a lower atrophy rate in the hippocampus volume. These associations are consistent across subgroups and are not altered by a genetic susceptibility to dementia. Additional extensive longitudinal studies involving diverse populations are required to validate the research findings and inform public health initiatives focused on enhancing brain health.

## Figures and Tables

**Figure 1 nutrients-16-01284-f001:**
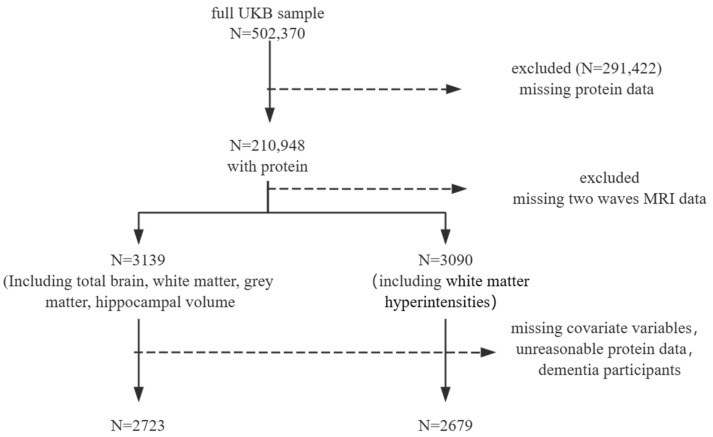
Participant inclusion.

**Figure 2 nutrients-16-01284-f002:**
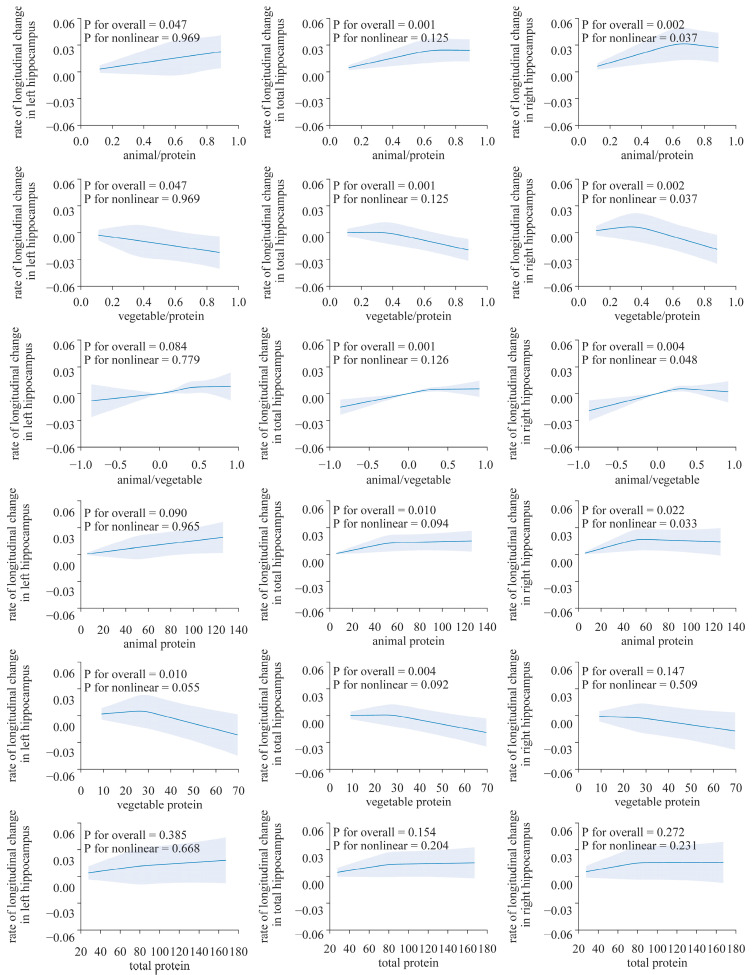
Nonlinear associations of dietary protein with the longitudinal change rate of the hippocampus volume. Using a restricted cubic spline regression model (N = 2723). The model was adjusted for age, sex, ethnicity, Townsend deprivation index, education level, physical activity, smoking, body weight status, total energy intake, baseline cancer, CVDs, hypertension and diabetes.

**Figure 3 nutrients-16-01284-f003:**
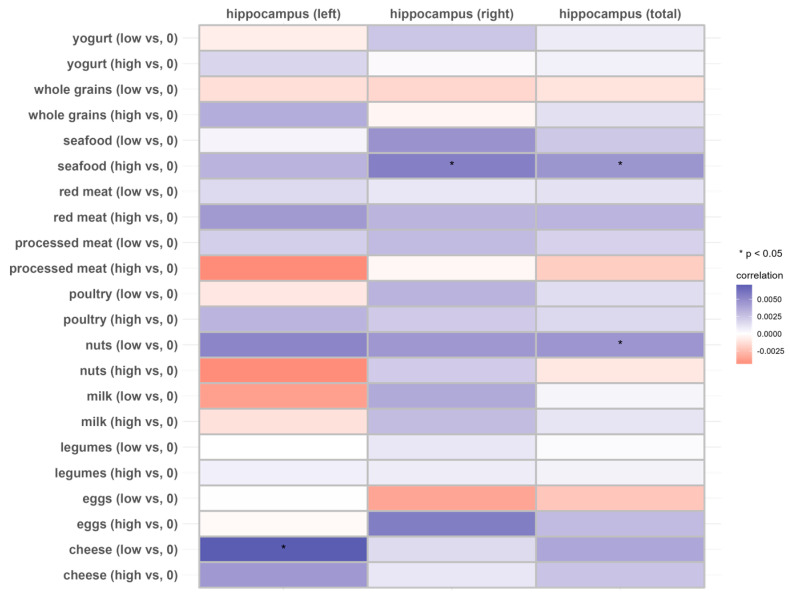
Associations of 11 dietary protein sources with the longitudinal change rate of the hippocampus volume (N = 2723). Beta coefficients and SE were calculated using general linear models adjusted for age, sex, ethnicity, Townsend deprivation index, education level, physical activity, smoking, body weight status, total energy intake, baseline cancer, CVDs, hypertension and diabetes.

**Table 1 nutrients-16-01284-t001:** Baseline characteristics of study participants (N = 2723).

	Total	Female	Male	*p*-Value
n	2723	1407	1316	
age, mean (SD)	52.66 (7.42)	51.65 (7.16)	53.74 (7.54)	<0.001
sex (%)				
female	1407 (51.7)	1407 (100.0)		
male	1316 (48.3)		1316 (100.0)	
MET (%)				0.376
low	474 (17.4)	233 (16.6)	241 (18.3)	
medium	1118 (41.1)	592 (42.1)	526 (40.0)	
high	1131 (41.5)	582 (41.4)	549 (41.7)	
TDI, mean (SD)	−1.99 (2.64)	−1.91 (2.68)	−2.07 (2.59)	0.110
smoke (%)				0.234
never	1742 (64.0)	915 (65.0)	827 (62.8)	
ever smoked	981 (36.0)	492 (35.0)	489 (37.2)	
race (%)				0.706
others	80 (2.9)	43 (3.1)	37 (2.8)	
white	2643 (97.1)	1364 (96.9)	1279 (97.2)	
drink (%)				0.101
never	56 (2.1)	35 (2.5)	21 (1.6)	
ever drunk	2667 (97.9)	1372 (97.5)	1295 (98.4)	
education (%)				0.133
below	1226 (45.0)	614 (43.6)	612 (46.5)	
college or above	1497 (55.0)	793 (56.4)	704 (53.5)	
BMI (%)				<0.001
underweight	16 (0.6)	13 (0.9)	3 (0.2)	
normal weight	1140 (41.9)	728 (51.7)	412 (31.3)	
overweight and obesity	1567 (57.5)	666 (47.3)	901 (68.5)	
cancer (%)	226 (8.3)	139 (9.9)	87 (6.6)	0.003
CVDs (%)	78 (2.9)	8 (0.6)	70 (5.3)	<0.001
hypertension (%)	529 (19.4)	163 (11.6)	366 (27.8)	<0.001
DM (%)	80 (2.9)	25 (1.8)	55 (4.2)	<0.001
animal protein, mean (SD)	53.03 (20.18)	50.71 (18.59)	55.50 (21.47)	<0.001
vegetable protein, mean (SD)	28.67 (9.65)	27.30 (9.11)	30.14 (10.00)	<0.001
proportion of animal protein, mean (SD)	0.64 (0.12)	0.64 (0.12)	0.64 (0.11)	0.844
proportion of vegetable protein, mean (SD)	0.36 (0.12)	0.36 (0.12)	0.36 (0.11)	0.844
animal/vegetable, mean (SD)	0.26 (0.24)	0.26 (0.25)	0.25 (0.23)	0.822
total protein, mean (SD)	81.70 (22.81)	78.02 (20.23)	85.64 (24.70)	<0.001

Data for continuous variables are presented as mean (SD). Data for categorical variables are presented as n (%). Abbreviations: MET, metabolic equivalent; BMI, body mass index; CVDs, cardiovascular diseases; SD, standard deviation; TDI, Townsend deprivation index; DM, diabetes mellitus.

**Table 2 nutrients-16-01284-t002:** The association between dietary protein intake and the longitudinal change rate of the hippocampus volume.

	Hippocampus (Left)		Hippocampus (Right)		Hippocampus (Total)	
	*β* (SE)	*p*	*β* (SE)	*p*	*β* (SE)	*p*
total protein						
model1	8.278 × 10^−6^ (5.342 × 10^−5^)	0.877	2.899 × 10^−5^ (4.785 × 10^−5^)	0.545	8.653 × 10^−6^ (3.606 × 10^−5^)	0.810
model2	9.956 × 10^−5^ (7.586 × 10^−5^)	0.190	7.375 × 10^−5^ (6.8 × 10^−5^)	0.268	7.547 × 10^−5^ (5.118 × 10^−5^)	0.141
model3	9.979 × 10^−5^ (7.592 × 10^−5^)	0.189	7.374 × 10^−5^ (6.803 × 10^−5^)	0.279	7.48 × 10^−5^ (5.1235 × 10^−5^)	0.144
animal/protein						
model1	2.581 × 10^−2^ (1009 × 10^−2^)	0.011 *	2.403 × 10^−2^ (9.034 × 10^−3^)	0.008 *	2.399 × 10^−2^ (6.7998 × 10^−3^)	0.001 *
model2	2.528 × 10^−2^ (1.021 × 10^−2^)	0.013 *	2.558 × 10^−2^ (9.148 × 10^−3^)	0.005 *	2.443 × 10^−2^ (6.881 × 10^−3^)	0.001 *
model3	2.524 × 10^−2^ (1.022 × 10^−2^)	0.014 *	2.544 × 10^−2^ (9.152 × 10^−3^)	0.005 *	2.435 × 10^−2^ (6.886 × 10^−3^)	0.001 *
vegetable/protein						
model1	−2.581 × 10^−2^ (1.009 × 10^−2^)	0.011 *	−2.403 × 10^−2^ (9.034 × 10^−3^)	0.008 *	−2.399 × 10^−2^ (6.71 × 10^−3^)	0.001 *
model2	−2.528 × 10^−2^ (1.021 × 10^−2^)	0.013 *	−2.558 × 10^−2^ (9.148 × 10^−3^)	0.005 *	−2.443 × 10^−2^ (6.881 × 10^−3^)	0.001 *
model3	−2.524 × 10^−2^ (1.022 × 10^−2^)	0.014 *	−2.544 × 10^−2^ (9.152 × 10^−3^)	0.005 *	−2.435 × 10^−2^ (6.886 × 10^−3^)	0.001 *
vegetable protein						
model1	−3.243 × 10^−4^ (1.257 × 10^−4^)	0.01 *	−1.644 × 10^−4^ (1.127 × 10^−4^)	0.145	−2.457 × 10^−4^ (8.479 × 10^−5^)	0.004 *
model2	−3.909 × 10^−4^ (1.646 × 10^−4^)	0.018 *	−2.731 × 10^−4^ (1.476 × 10^−4^)	0.065	−3.194 × 10^−4^ (1.111 × 10^−4^)	0.004 *
model3	−3.901 × 10^−4^ (1.648 × 10^−4^)	0.018 *	−2.724 × 10^−4^ (1477 × 10^−4^)	0.065	−3.190 × 10^−4^ (1.112 × 10^−4^)	0.004 *
animal protein						
model1	8.415 × 10^−5^ (5.996 × 10^−5^)	0.161	−1.644 × 10^−5^ (1.127 × 10^−4^)	0.169	6.676 × 10^−5^ (4.046 × 10^−5^)	0.099
model2	1.522 × 10^−4^ (6.924 × 10^−5^)	0.028 *	1.111 × 10^−4^ (6.208 × 10^−5^)	0.074	1.194 × 10^−4^ (4.671 × 10^−5^)	0.011 *
model3	1.522 × 10^−4^ (6.929 × 10^−5^)	0.028 *	1.096 × 10^−4^ (6.211 × 10^−5^)	0.078	1.188 × 10^−4^ (4.674 × 10^−5^)	0.011 *
animal/vegetable						
model1	1.281 × 10^−2^ (5 × 10^−3^)	0.01 *	1.127 × 10^−2^ (4.451 × 10^−3^)	0.011 *	1.16 × 10^−2^ (3.3 × 10^−3^)	0.001 *
model2	1.251 × 10^−2^ (5.03 × 10^−3^)	0.01 *	1.199 × 10^−2^ (4.508 × 10^−3^)	0.008 *	1.176 × 10^−2^ (3.391 × 10^−3^)	0.001 *
model3	1.249 × 10^−2^ (5.033 × 10^−3^)	0.01 *	1.193 × 10^−2^ (4.509 × 10^−3^)	0.008 *	1.173 × 10^−2^ (3.393 × 10^−3^)	0.001 *

*β*: beta coefficients; SE: standard error; *, *p* < 0.05, statistical significance. *β* and SE were calculated through multiple linear regression modeling and were adjusted for multiple factors. Model 1 was adjusted for age and sex. Model 2 was based on model 1 and additionally adjusted for the Townsend deprivation index, total energy intake, education level, physical activity, smoking, alcohol intake, race and body weight status. Model 3 was based on model 2 and further adjusted for baseline cancer, CVDs, hypertension and diabetes.

## Data Availability

The data that support the findings of this study are available from UK Biobank (http://www.ukbiobank.ac.uk/about-biobank-uk/ accessed on 15 February 2023). Restrictions apply to the availability of these data, which were used under license for the current study (Project ID: 95715). Data are available for bona fide researchers upon application to UK Biobank.

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
