# Peer review of "The Association between Dietary Protein Intake and Sources and the Rate of Longitudinal Changes in Brain Structure"

_nutrients, 2024, doi:10.3390/nu16091284_

Round 1

Reviewer 1 Report

Comments and Suggestions for Authors

The manuscript (nutrients-2950722) examined the correlation between dietary protein intake, in particular that of animal or seafood, and MRI measures of brain volume changes over time.  Higher animal protein in diet, particularly from seafood, was found to correlate negatively with the rate of change in hippocampal volume. It was concluded that a moderate increase in the intake and proportion of animal protein in the diet, especially from seafood, is associated with a lower longitudinal change rate in hippocampus volume.

Overall, the finding of the manuscript appears to be interesting.  I have a few comments for the authors:

1.      The overall span of the two image acquisitions should be indicated in the abstract;

2.      What is the overall change in hippocampal volume between the two image acquisitions, in terms of absolute volume, or percentage change? And how is it compared to the literature? If I understand it correctly, from Figure 3, most of the subjects had an increase in hippocampal volume between the two imaging sessions. I suggest the authors reverse their definition of volume change, i.e., (2019+ - 2014+)/2014+, which is more straightforward for readers.

3.      Basically the association with the proportion of vegetable protein is the opposite from that of animal protein; thus it is redundant to have both in the results. In Figure 3, I don’t understand why the y-axis values appear to be reversed between vegetable and animal protein. Please double-check.

4.      Statistics should be indicated in Table 2 so that it is clear to the readers;

5.      In Figure 2, why the association with absolute protein was not depicted?

6.      The introduction or discussion can expand a bit to summarize the overall trend in the research on diet and brain changes, not necessarily consistent with findings of the current study.

Author Response

Response to Reviewer 1 Comments

1. Summary

Thank you very much for taking the time to review this manuscript. Please find the detailed responses below and the corresponding revisions highlighted in the re-submitted files.

2. Questions for General Evaluation

Reviewer’s Evaluation

Response and Revisions

Does the introduction provide sufficient background and include all relevant references?

Can be improved

Are all the cited references relevant to the research?

Yes

Is the research design appropriate?

Yes

Are the methods adequately described?

Yes

Are the results clearly presented?

Can be improved

Are the conclusions supported by the results?

Yes

3. Point-by-point response to Comments and Suggestions for Authors

Comments 1: The overall span of the two image acquisitions should be indicated in the abstract;

Response 1: Thank you for pointing this out. We agree with this comment. Therefore, we added description on the overall span between two consecutive image acquisitions in the abstract. (Highlighted in Lines on Page 1: 17-18)

Comments 2: What is the overall change in hippocampal volume between the two image acquisitions, in terms of absolute volume, or percentage change? And how is it compared to the literature? If I understand it correctly, from Figure 3, most of the subjects had an increase in hippocampal volume between the two imaging sessions. I suggest the authors reverse their definition of volume change, i.e., (2019+ - 2014+)/2014+, which is more straightforward for readers.

Response 2: Thanks very much for your comments. In our study, there was an average hippocampal volume reduction of 1.6% between the two waves of brain imaging. In previous studies, the hippocampus shows annual atrophy rates from 0.79% to 2.0%[1-3]. To make the results more intuitive, we redefined the longitudinal change rate as (2019+ - 2014+)/2014+ as you suggested and recalculated the results accordingly. Modifications have been made to the manuscript, figures, and tables accordingly. (Highlighted in Lines on Page 3: 97-101)

[1] Fischl B, Salat DH, Busa E, Albert M, Dieterich M, Haselgrove C, et al. Whole brain segmentation: automated labeling of neuroanatomical structures in the human brain. Neuron. 2002;33(3):341-55.

[2] Fjell AM, Walhovd KB. Structural brain changes in aging: courses, causes and cognitive consequences. Reviews in the neurosciences. 2010;21(3):187-221.

[3] Fjell AM, Walhovd KB, Fennema-Notestine C, McEvoy LK, Hagler DJ, Holland D, et al. One-year brain atrophy evident in healthy aging. The Journal of neuroscience : the official journal of the Society for Neuroscience. 2009;29(48):15223-31.

Comments 3: Basically, the association with the proportion of vegetable protein is the opposite from that of animal protein; thus, it is redundant to have both in the results. In Figure 3, I don’t understand why the y-axis values appear to be reversed between vegetable and animal protein. Please double-check.

Response 3: Thanks very much for your advice. We have removed redundant descriptions of the results. We also re-examined the results of the restricted cubic spline plots in the original Figure 3 (now Figure 2) and made adjustments to the axes of the graphs. The y-axis represents the longitudinal change rates in different hippocampal regions, while the x-axis represents the relative and absolute indicators of dietary protein.

Comments 4: Statistics should be indicated in Table 2 so that it is clear to the readers.

Response 4: Thank you very much for your comments. We added a description of the statistics in Table 2. (Highlighted in Table 2)

Comments 5: In Figure 2, why the association with absolute protein was not depicted?

Response 5: Thank you very much for your advice. We rechecked our results and found that due to the magnitude issue of confidence intervals, the results for absolute protein could not be fully displayed in original Figure 2. The 95% confidence interval for the association between total protein and the left hippocampus was -2.486e-04 to 4.901e-05, while the 95% confidence interval for the association between animal protein as a proportion of total protein and the left hippocampus was -4.527e-02 to -5.209e-03. The length difference between the two intervals was significant, making it impossible to display the confidence interval for absolute protein in the figure. Considering that the content shown in Table 2 and Figure 2 was almost the same, we therefore removed the original Figure 2 and only retained Table 2 to display the results.

Comments 6: The introduction or discussion can expand a bit to summarize the overall trend in the research on diet and brain changes, not necessarily consistent with findings of the current study.

Response 6: Thank you very much for your advice. We consulted relevant literature and further supplemented and improved the discussion section. (Highlighted in Lines on Page 11: 259-270)

Reviewer 2 Report

Comments and Suggestions for Authors

In this manuscript, the authors describe the results of an association study between dietary protein intake and macroscopic changes in the hippocampus. Interestingly, they found a significant correlation between the changes observed in hippocampal volume and protein intake, specifically, that animal proteins seem to be protective. This could greatly impact our understanding of the effects of dietary components on brain health if it can be confirmed in exploratory studies.

However, some concerns need to be addressed to make the presented results more understandable and reliable.

First, the authors should explain why the neurological diseases were not defined as exclusion criteria or included as covariates. Seizures, for example, were shown to affect hippocampal volume in a 2- to 3-year follow-up period (10.1212/01.WNL.0000148643.36513.2A or 10.1148/rg.210153). Moreover, infectious diseases (various types of encephalitis) should have also been assessed. 

Second, psychiatric conditions are also associated with smaller hippocampal volume than healthy controls (10.1177/2470547020906799). In addition, a recent study revealed that "Individuals who excluded meat from their diet had a higher prevalence of depressive episodes. (10.1016/j.jad.2022.09.05) ". Consequently, lower animal protein intake could lead to psychiatric conditions that were already linked to smaller hippocampal volume. The authors should comment further on this.

Third, there is some evidence that estrogens affect hippocampal volume. Previous oral contraceptive use has been linked with hippocampal enlargement (10.1038/s41598-019-47446-4), and abrupt estradiol withdrawal (maybe menopause) could reduce hippocampal volume (10.1016/j.yhbeh.2022.105234). 

The authors should check whether the above mentioned conditions might influence the results of their study or these should be acknowledged as further limitations of the study. 

Comments on the Quality of English Language

Minor changes are required.

Author Response

Response to Reviewer 2 Comments

1. Summary

Thank you very much for taking the time to review this manuscript. Please find the detailed responses below and the corresponding revisions highlighted in the re-submitted files.

2. Questions for General Evaluation

Reviewer’s Evaluation

Response and Revisions

Does the introduction provide sufficient background and include all relevant references?

Can be improved

Are all the cited references relevant to the research?

Must be improved

Is the research design appropriate?

Can be improved

Are the methods adequately described?

Yes

Are the results clearly presented?

Yes

Are the conclusions supported by the results?

Can be improved

3. Point-by-point response to Comments and Suggestions for Authors

Comments 1: First, the authors should explain why the neurological diseases were not defined as exclusion criteria or included as covariates. Seizures, for example, were shown to affect hippocampal volume in a 2- to 3-year follow-up period (10.1212/01.WNL.0000148643.36513.2A or 10.1148/rg.210153). Moreover, infectious diseases (various types of encephalitis) should have also been assessed. 

Response 1: Thank you for pointing this out. Existing research has indicated that longitudinal changes in brain structural markers can reflect alterations in the progression of neurodegenerative diseases to some extent [1-4]. We initially excluded participants with baseline dementia. However, after excluding missing covariates, dementia was not present at baseline, so we omitted it in Figure 1. Therefore, we re-drew the flowchart (Figure 1) and supplemented it in the results.

Considering the influence of neuropsychiatric disorders[5] on the hippocampus, we further performed a sensitivity analysis by excluding baseline neuropsychiatric disorders (inculding depressive, epilepsy, and encephalitis). The results showed no changes, indicting that the results were stable and robust. These were supplemented in the method and discussion (Highlighted in Lines on Page 3: 139-144; Page 11: 259-270; Supplemental Table 11).

[1] Frisoni GB, Fox NC, Jack CR, Jr., Scheltens P, Thompson PM. The clinical use of structural MRI in Alzheimer disease. Nature reviews Neurology. 2010;6(2):67-77.

[2] Dickerson BC, Bakkour A, Salat DH, Feczko E, Pacheco J, Greve DN, et al. The cortical signature of Alzheimer's disease: regionally specific cortical thinning relates to symptom severity in very mild to mild AD dementia and is detectable in asymptomatic amyloid-positive individuals. Cerebral cortex (New York, NY : 1991). 2009;19(3):497-510.

[3] Dubois B, Hampel H, Feldman HH, Scheltens P, Aisen P, Andrieu S, et al. Preclinical Alzheimer's disease: Definition, natural history, and diagnostic criteria. Alzheimer's & dementia : the journal of the Alzheimer's Association. 2016;12(3):292-323.

[4] Elias MF, Beiser A, Wolf PA, Au R, White RF, D'Agostino RB. The preclinical phase of alzheimer disease: A 22-year prospective study of the Framingham Cohort. Archives of neurology. 2000;57(6):808-13.

[5] Gupta R, Advani D, Yadav D, Ambasta RK, Kumar P. Dissecting the Relationship Between Neuropsychiatric and Neurodegenerative Disorders. Molecular neurobiology. 2023;60(11):6476-529.

Comments 2: Second, psychiatric conditions are also associated with smaller hippocampal volume than healthy controls (10.1177/2470547020906799). In addition, a recent study revealed that "Individuals who excluded meat from their diet had a higher prevalence of depressive episodes. (10.1016/j.jad.2022.09.05) ". Consequently, lower animal protein intake could lead to psychiatric conditions that were already linked to smaller hippocampal volume. The authors should comment further on this.

Response 2: Thank you very much for your advice. We further excluded baseline neuropsychiatric disorders (including depressive, epilepsy, and encephalitis) in sensitivity analysis. The association still existed. These were also supplemented in the method and discussion (Highlighted in Lines on Page 3: 139-144; Page 11: 259-270; Supplemental Table 10).

Comments 3: Third, there is some evidence that estrogens affect hippocampal volume. Previous oral contraceptive use has been linked with hippocampal enlargement (10.1038/s41598-019-47446-4), and abrupt estradiol withdrawal (maybe menopause) could reduce hippocampal volume (10.1016/j.yhbeh.2022.105234). 

Response 3: Thanks very much for your advice. We additionally adjusted for oral contraceptive use in female in sensitivity analysis. The main results showed no changes. These were also supplemented in the manuscript. (Highlighted in Lines on Page 3: 139-144; Supplemental Table 11)

Round 2

Reviewer 2 Report

Comments and Suggestions for Authors

The authors addressed all the reviewer's concerns and included in their analysis pathological risk factors that affect hippocampal volume.

As a result of the revisions made, the manuscript was improved and can now be accepted for publication.